# Influences of the Heating and Cooling Rates on the Dissolution and Precipitation Behavior of a Nickel-Based Single-Crystal Superalloy

**Xiao-Yan Wang [1,2,\*], Zhi-Xun Wen [1], Hao Cheng [1], Shu-Ning Gu [1] and Guang-Xian Lu [1]**

[1] School of Mechanics, Civil Engineering and Architecture, Northwestern Polytechnical University, Xi'an 710072, China; zxwen@nwpu.edu.cn (Z.-X.W.); fanran@mail.nwpu.edu.cn (H.C.); gushuning@mail.nwpu.edu.cn (S.-N.G.); luguangxian@mail.nwpu.edu.cn (G.-X.L.)

[2] School of Science, Xi'an Shiyou University, Xi'an 710065, China

\* Correspondence: shiyouwxy@126.com

**Abstract:** The effects of the heating rate before solution treatment, and the cooling rate after solution treatment on the morphological distribution and evolution of the precipitation phase of nickel-based single crystal superalloy were studied. The dissolution, precipitation, and growth of the precipitation phase and the matrix phase during heat treatment were analyzed by the means of high-power scanning electron microscopy. The results show that the morphology of the precipitated phase has nothing to do with the distribution of the precipitated phase and the heating rate in the heating process, but the cooling rate in the cooling process affects the shape, size, and distribution of the precipitated phase. The faster the cooling rate, the smaller the precipitated phase is, the more irregular the shape is, the smaller the equivalent edge length is, and the smaller the channel width of the matrix phase is.

**Keywords:** nickel-based single crystal alloy; heat treatment; heating and cooling rate; precipitation and dissolution of γ′ phase

## 1. Introduction

Nickel-based single crystals have gradually become the preferred material for aero-engine turbine blades because of their excellent mechanical properties at high temperature, especially creep properties, oxidation resistance, corrosion resistance, high toughness, and good processing plasticity [1]. Good high-temperature mechanical properties were obtained by precipitation strengthening. The molecular structure of $Ni_3$ (Al, Ti) as precipitation strengthening phase was a primitive cubic crystal structure Pm3m. The macroscopic structure was cubic, and the strengthening phase was surrounded by continuous solid solvent matrix phase, which presented a chessboard-like compact and regular arrangement [2]. It has been found that the basic characteristics of the strengthened phase, such as shape, size, distribution, and volume fraction, have an important effect on the mechanical properties of superalloys [3–6].

Many scholars have done a series of work on the dissolution and precipitation behavior of precipitation phases of Ni-based alloys at high temperature. The microstructural evolution of precipitation phases has been discussed. The morphological changes and distribution of the precipitation of γ phases during continuous cooling after solid solution treatment have been studied. Good results have been obtained in the recovery heat treatment system of the alloys. Wang [7] mainly studied the effect of solution temperature and holding time on the γ-phase dissolution behavior of IN100 and DS Rene 125 alloys during solution treatment, and explored the dynamic factors of precipitation phase dissolution behavior during solution treatment. The results show that the

environmental temperature has a greater influence on the dissolution of the precipitation phase precipitation than the holding time. At the same time, the morphology of the precipitation phase is strongly influenced by the elastic-strain energy in the structure, which is related to the lattice mismatch between the precipitation phase and the $\gamma$ phase. Grosdidier et al. [8] studied the precipitation and dissolution of the $\gamma$-phase in nickel-based alloys AM1 and CMSX-2 during heat treatment, discussed the microstructural evolution of the precipitation phase just after precipitation, and proposed that the precipitation phase would undergo deformation from spherical to cubic to dendritic from nucleation to growth. Gao et al. [9] studied the effect of cooling rate on the microstructural evolution and mechanical properties of the Ni-based superalloy Rene 80 during the cooling process after solution treatment. It was found that the higher the cooling rate, the smaller the precipitated precipitation phases during solidification, and the faster the coarsening of the precipitation phases in the subsequent aging process. After many mechanical properties tests on alloys with different cooling rates, it was concluded that air cooling is the most suitable cooling rate for Rene 80 alloy in the cooling stage of heat treatment. Jackson et al. [10] studied the directionally solidified superalloy (DS) Mar-M200+HF. It was found that a higher solution temperature could dissolve more coarsened $\gamma'$ phases and $\gamma$-$\gamma'$-eutectic structures, and more high-quality gamma-precipitation phases were precipitated during subsequent cooling and aging. Increasing the volume fraction of high-quality precipitation phases could greatly improve the creep properties of the alloy, When the volume fraction of the strengthening phase increases from 30% to 45%, the creep fracture life of the material increases to three times as long as the original lifetime. Li et al. [11] found in the study of FGH4096 blade materials that the cooling rate after solution treatment determined the morphology and distribution of the precipitated precipitation phases. A faster cooling rate could make the precipitated precipitation phases have better morphology and a narrower channel of matrix phase, which improved the precipitation effect of Orowan [12–14], and effectively hindered dislocation movement in matrix phase; thus, the material could be improved. The creep properties were improved at 704 °C/690 MPa. At present, DD6 is widely used in the hot-end components of aero-engines. Besides, plastic deformation caused by load, there is no grain boundary, and the evolution law of the microstructure of the precipitation phase and the matrix phase during creep and after the heat treatment process is different from that of general superalloys. At high temperature, the microstructure morphology of DD6 will change to a certain extent, which will affect the performance of the blades [15–18]. The study of the dissolution and precipitation behavior of the $\gamma'$ phase at high temperature has important practical engineering significance.

Starting from different stages of the heat treatment system, we focus on many factors, such as the heating rate in the solution treatment stage, the cooling rate in the cooling stage after solution treatment, and the aging temperature and time in the aging stage. By designing different experimental conditions, the effects of various factors (heating rate and cooling rate) on the dissolution and precipitation behavior of the $\gamma'$ phase and the number, size, morphology, and distribution of the final precipitates in the heat treatment system were studied.

## 2. Materials and Methods

### 2.1. Material

A nickel-based single crystal was used as the experimental material and standard two-phase alloys, which were the $\gamma$ phase and $\gamma'$ phase, respectively. The $\gamma'$ phase is cubic, and it is arranged in an orderly manner, with an equivalent side-length of about 530 nm. Figure 1 shows the microstructure of the dendrite cadre site of the second-generation nickel-based single crystal alloy DD6 after standard heat treatment (1290 °C × 1 h + 1300 °C × 2 h + 1315 °C × 4 h/AC + 1120 °C × 4 h/AC + 870 °C × 32 h/AC, AC means air-cooling). The chemical composition of the material is listed in Table 1.

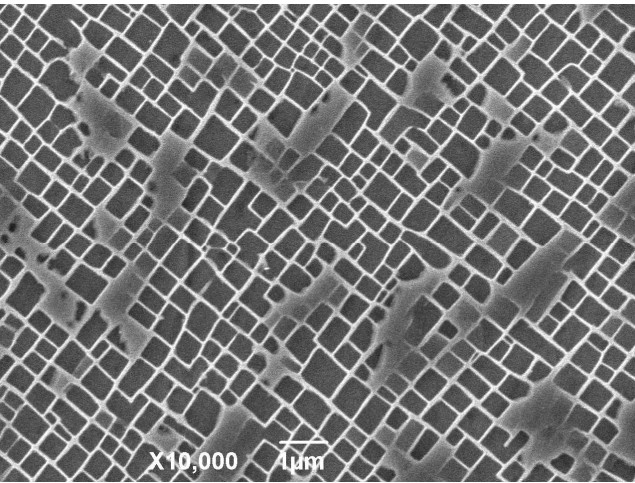

**Figure 1.** Microstructure of the nickel-based single crystal.

**Table 1.** Chemical constituents of the nickel-based single crystal (mass fraction%).

| Phase | Al | Cr | Co | Ni | Mo | Ta | W | Re |
|-------|------|------|-------|-------|------|------|------|------|
| $\gamma'$ phase | 5.18 | 4.52 | 9.21 | 61.33 | 0.79 | 7.16 | 7.48 | 2.09 |
| $\gamma$ phase | 4.64 | 5.21 | 10.57 | 60.67 | 1.73 | 5.31 | 8.74 | 2.13 |

### 2.2. Sample Test

In order to facilitate the test, cubic specimens were used to prepare the samples by a metallographic sample preparation procedure, because this paper focuses on the changes of the microstructures of the specimens during heat treatment without external loads, and it does not involve the study of the macro-mechanical properties. The DD6 material was cut into 6 mm × 5 mm × 3 mm metal blocks, and the 6 mm × 5 mm surface was in the direction of [001] along the 6 mm-length direction. After the experiment, one of the 6 mm × 5 mm planes was polished on sandpaper with grain size of 800 mesh, 1200 mesh, 1500 mesh, and 2000 mesh, and then polished onto the polishing cloth. The planar finish should meet the requirement of no scratches under a 50× magnifying glass. A chemical etching solution was prepared with $HNO_3$, HF solution, and glycerol, in certain proportions. The etching droplets were washed with clean water after 3–5 s on the polished surface, and the operation was repeated 3–4 times until the surface was slightly blue, and then the etching operation could be completed. JSM-6390A scanning electron microscope was used to observe the microstructures and distributions of the strengthened and matrix phases on the polished surface.

### 3. Experimental Process

The heat treatment process adopted in this experiment can be roughly divided into three distinct stages: solid solution treatment, cooling after solid solution treatment, and aging treatment. The effects of heating rate, cooling rate after solution treatment, aging temperature, and time, on the dissolution and precipitation behaviors of the precipitation phases in the DD6 single crystal alloy, and the morphological distribution of the precipitation phases were studied. All experiments were repeated at least three times to ensure the reliability of the results.

The entire heat treatment test was carried out in a special high-temperature furnace. The device can set up with several programmed sections to control the temperature rise and fall at a specified rate, and to accurately keep the temperature constant.

### 3.1. Solution Treatment Stage

The solid solution treatment stage is divided into the heating process and the heat preservation process. The effects of temperature and holding time on the dissolution behaviors of materials have

been studied in a large number of studies. In order to explore the influence of heating rate on the final precipitation morphology distribution during the heating process, the following experimental process was designed: a solution temperature of 20 °C higher than the standard solution temperature of heat treatment (i.e., 1335 °C) was taken as the solution temperature of the new heat treatment system. The samples were heated to 1335 °C at different heating rates, which were: ① rapid heating, ② 0.18 °C/s, and ③ 0.09 °C/s, respectively. The test process is shown in Figure 2.

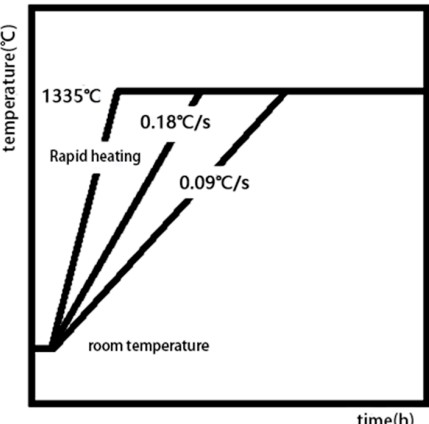

**Figure 2.** Heating process.

Because the temperature in the early stage of the heating process is very low and far below the dissolution temperature of the precipitation phase, many precipitation phases remain undissolved at the end of the heating process, and because the different heating rates lead to different amounts of dissolution for the precipitation phase at the end of the heating process, a uniform holding temperature of the sample of 1335 °C for 4 h was chosen, to ensure the complete dissolution of the phase, and then air cooling was carried out.

*3.2. The Cooling Stage*

In the solid-solution treatment stage, the precipitation phase gradually dissolved in the matrix phase to form a single-phase solid solution, after holding for a long time at high temperature. As the solubility of the precipitation phase in the matrix phase decreased with the decrease of temperature, when the solid solution treatment was finished and the cooling stage began, with the decrease of temperature, the single-phase solid solution gradually began to precipitate the precipitation phase, because of its supersaturation. According to the study of Grosdidier et al. [8], the precipitation behavior of the precipitation phase was affected by the different cooling rate. In order to study the effect of cooling rate on the shape, size, and distribution of the precipitated phases, seven different cooling rates were adopted under the same heating rate, solution temperature, and treatment time: (a) 0.15 °C/s, (b) 0.25 °C/s, (c) 0.4 °C/s, (d) 0.5 °C/s, (e) 0.6 °C/s, (f) air-cooled, and (g) water-cooled to room temperature at a constant rate. The experimental flow chart of the effect of different cooling rates on precipitation is shown in Figure 3.

*3.3. Aging Stage*

The aging process of single-crystal alloys is the growth and re-precipitation of precipitation phases, which essentially depends on the diffusion of solute atoms represented by Al atoms. Therefore, the aging temperature and time control the formation and morphology distribution of the precipitation phases. It has been proved that the volume fraction, cubic degree, and regularity of phase arrangement of high-quality precipitation phases have great influences on the creep and fatigue properties of single-crystal materials, so that aging treatment plays an important role in improving the mechanical properties of single-crystal materials. In order to study the influence of aging temperature and aging

time on the shape, size, and distribution of precipitates in different aging treatment systems, five different aging treatment systems, as listed in Table 2, are adopted. The specific experimental process is shown in Figure 4.

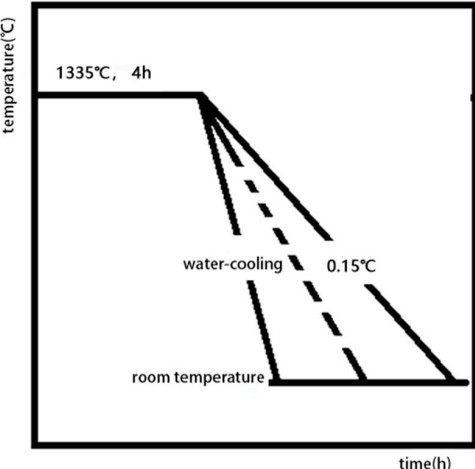

**Figure 3.** Cooling process.

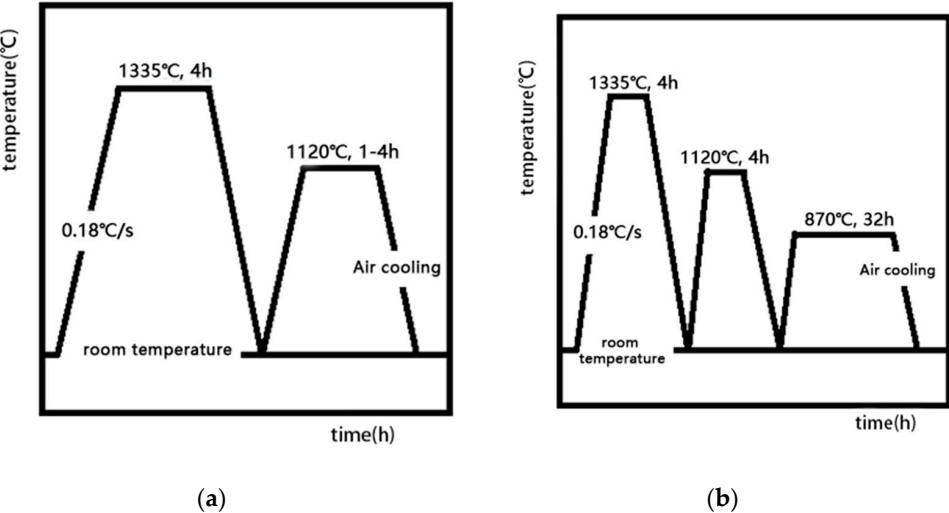

**Figure 4.** Aging process diagrams (**a**) The aging treatment system: ①–④; (**b**) The aging treatment system: ⑤.

**Table 2.** Treatment systems.

| Number | Solution Treatment | Aging Treatment |
|---|---|---|
| ① | 1335 °C/4 h | 1120 °C/1 h |
| ② | 1335 °C/4 h | 1120 °C/2 h |
| ③ | 1335 °C/4 h | 1120 °C/3 h |
| ④ | 1335 °C/4 h | 1120 °C/4 h |
| ⑤ | 1335 °C/4 h | 1120 °C/4 h + 870 °C/32 h |

## 4. Experimental Results and Analysis

### 4.1. Heating Rate

Three different heating rates were used to polish and corrode the specimens. The samples were heated from room temperature to the target temperature at different heating rates, and after full solution treatment, the temperature was reduced to room temperature at the same cooling

rate. The microstructures of the precipitates were observed under high-power electron microscopy. The results are as follows: Figure 5. The average equivalent edge length of precipitated phase after cooling was 80 nm (Figure 5a), at rapid heating, 82 nm (Figure 5b) at the heating rate of 0.18 °C/s, and 79 nm (Figure 5c) at the heating rate of 0.09 °C/s. The average equivalent side lengths of the precipitated phases at different rates are shown in Figure 5c.

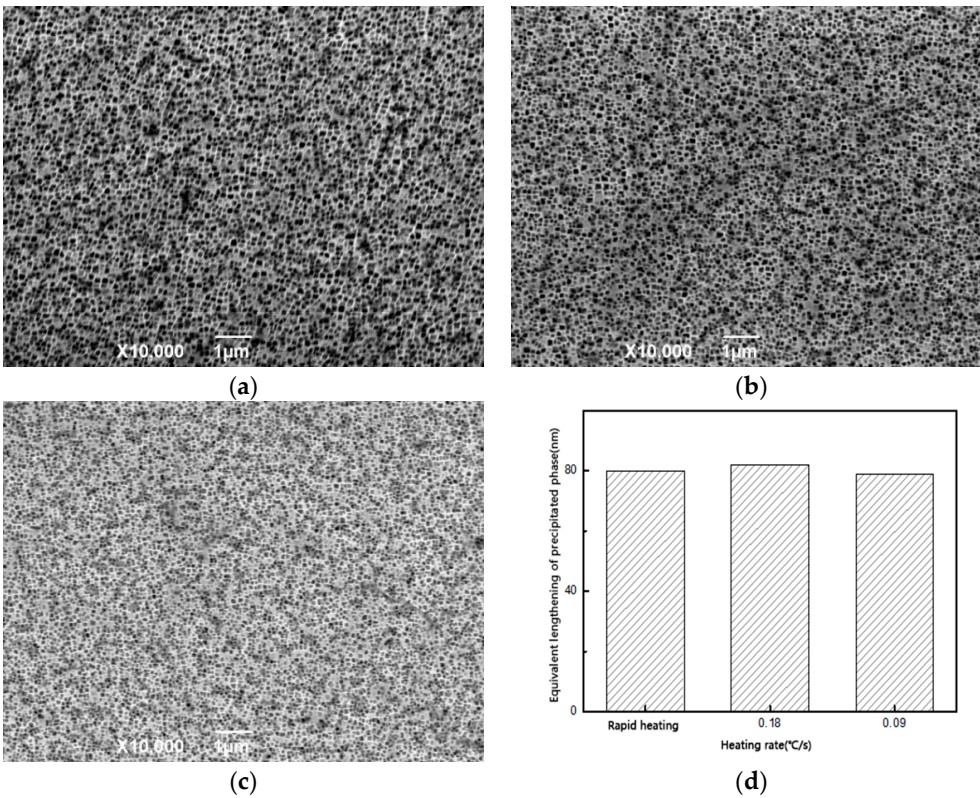

**Figure 5.** Microstructure of the γ/γ′ phase after cooling by solution treatment at different temperature rising rates (**a**) Rapid heating, (**b**) 0.18 °C/s, (**c**) 0.09 °C/s, (**d**) Average equivalent side length of precipitated phase.

To determine the effects of three heating rate conditions on the distribution of elements in the channel of the precipitated phase and matrix phase obtained after cooling, the surface of the sample was analyzed by EDS (Energy Dispersive Spectrometer). The results are listed in Tables 3 and 4.

**Table 3.** Chemical constituents (mass fraction%) of the precipitation phases at three heating rates.

| Heating Rate | Al | Cr | Co | Ni | Mo | Ta | W | Re |
|---|---|---|---|---|---|---|---|---|
| Rapid heating | 4.55 | 4.57 | 9.44 | 60.60 | 2.12 | 5.98 | 9.15 | 2.59 |
| 0.18 °C/s | 4.61 | 4.59 | 9.25 | 60.93 | 2.57 | 5.83 | 8.41 | 2.81 |
| 0.09 °C/s | 4.69 | 4.60 | 9.30 | 60.14 | 2.38 | 6.34 | 9.19 | 2.36 |

**Table 4.** Chemical composition (mass fraction%) of matrix phase at three heating rates.

| Heating Rate | Al | Cr | Co | Ni | Mo | Ta | W | Re |
|---|---|---|---|---|---|---|---|---|
| Rapid heating | 5.30 | 4.23 | 8.93 | 59.59 | 3.33 | 6.03 | 7.87 | 2.75 |
| 0.18 °C/s | 5.71 | 4.71 | 9.85 | 58.49 | 2.25 | 6.24 | 8.20 | 2.56 |
| 0.09 °C/s | 5.18 | 4.66 | 9.49 | 59.21 | 2.67 | 6.32 | 8.36 | 2.13 |

By comparing the phase diagrams obtained under the three heating rates in Figure 5, it can be found that the morphological distributions of the precipitated phases were very similar,

the cross-sections were mostly round, and a few of them showed trends of quadrangle preliminarily. The equivalent side length of the precipitated phase was very close, approximating 80 nm, and the width of the matrix channel between the phases was less than 10 nm.

From the comparison of the results of the energy spectrum analysis, it can be seen that the samples were heated to solution temperature at three different heating rates, and treated with solid solution for a sufficient time (4 h). The samples were lowered to room temperature at the same cooling rate (air cooling). The contents of the main elements in the precipitation phase were basically the same. The contents of Al, Cr, Co, Mo, Ta, W, Re, and other elements in the precipitation phase were basically the same under three conditions. Being closely approximate, it can be seen that the heating rate conditions had little effect on the migration of the elements.

It can be concluded that the morphology of the enhanced phase after precipitation has nothing to do with the distribution and the heating rate during the heating process.

### 4.2. Cooling Rate

In order to study the effect of the cooling rate on the precipitation behavior of the precipitation phase in the cooling stage after solid solution treatment during heat treatment, DD6 samples were heated to 1335 °C and kept for 4 h, so that all phases in the original sample were completely dissolved into the matrix phase to form a uniform solid solution. After cooling operations, seven different cooling rates were taken, namely (a) 0.15 °C/s, (b) 0.25 °C/s, (c) 0.4 °C/s, (d) 0.5 °C/s, (e) 0.6 °C/s, and (f) air-cooling and (g) water-cooling; the samples were then cooled to room temperature. Figure 6 shows the microstructures of the precipitated precipitation phases in the sample matrix at various cooling rates. The average equivalent edge lengths of the precipitates in Figure 6 were (a) 375 nm, (b) 273 nm, (c) 187 nm, (d) 169 nm, (e) 132 nm, (f) 86 nm, (g) 12 nm, respectively.

It can be seen from Figure 6 that the shape, size, and distribution of the precipitates depended on the cooling rate during cooling. When the cooling rate was more than 300 °C/s, i.e., water-cooled conditions, the precipitate phase was very small, its shape was irregular, point-like and circular, its equivalent edge length was less than 20 nm, and the width of the matrix phase channel was less than 1 nm; the matrix phase channel was arranged closely and in a disorderly arrangement, and the channel was very narrow, with more fine secondary precipitates. It is difficult to clearly display the microstructures under existing electron microscopy conditions. When the cooling rate was greater than 50 °C/s, i.e., air cooling, most of the precipitated phases were circular in shape, and some of them had begun to grow in a quadrangular direction, showing a trend towards a square transition. The equivalent side length was about 86 nm, the width of the matrix phase channel was approximately 8 nm, and the fine secondary precipitated phases in the channel were greatly reduced. Compared with the water cooling condition, the number of precipitated phases was reduced and the volume was increased. The air cooling rate was still very fast, the precipitation time was very short, and the time available for a phase growth and a merger was very small. When the cooling rate was between 0.4 and 0.6 °C/s, the precipitation phase presented an irregular square shape, and the circular precipitation phase decreased or even disappeared. Compared with water cooling and air cooling, the cooling rate decreased greatly. The precipitation phase had a certain amount of time to grow up and merge with each other. The circular phase gradually grew into a rectangular outline, the equivalent edge reached 180 nm, and the distribution gradually became regular, and the matrix was interconnected. The channel became obvious, the width of the channel was about 15 nm, and the secondary precipitation phase in the channel decreased greatly. When the cooling rate was less than 0.25 °C/s, and especially less than 0.15 °C/s, it can be seen from the figure that the precipitated phase presented a regular block shape and larger volume, the equivalent side length was about 300 nm, in a more orderly arrangement, and the width of the channel of the matrix phase was about 30 nm. The shape of the precipitated phase was slightly deformed, and there was a tendency to form a butterfly-like structure with four precipitation phases.

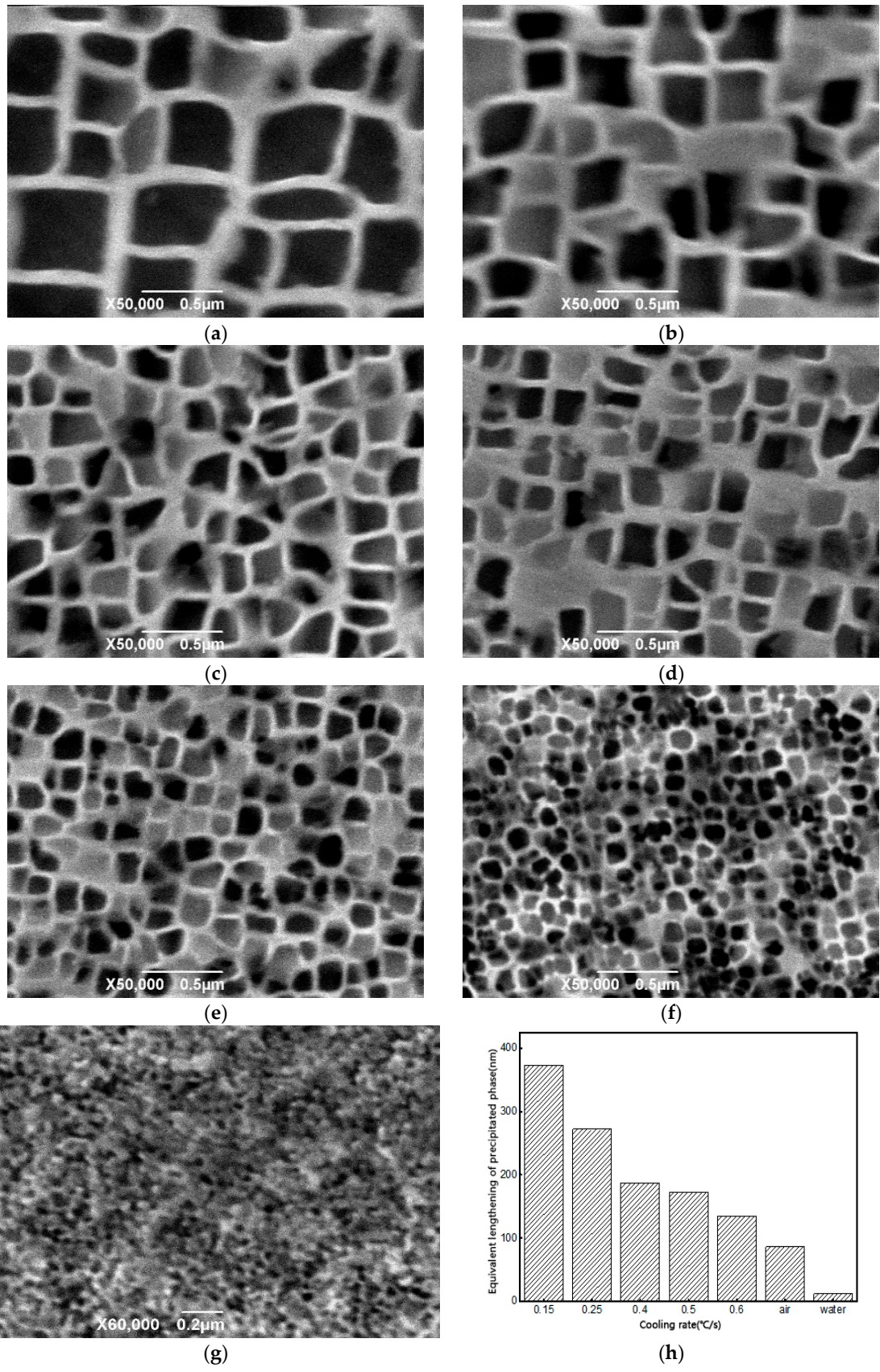

**Figure 6.** Microstructure of materials obtained by different cooling rates (**a**) 0.15 °C/s; (**b**) 0.25 °C/s; (**c**) 0.4 °C/s; (**d**) 0.5 °C/s; (**e**) 0.6 °C/s; (**f**) Air-cooling; (**g**) Water-cooling; (**h**) Average equivalent side length of the precipitated phase at different cooling rates.

### 4.3. Aging Treatment

After the sample was heated to 1335 °C and held for 4 h, the air-cooling method was used to cool the sample to room temperature, and then the following aging treatments were carried out respectively: holding (a) 0 h, (b) 1 h, (c) 2 h, (d) 3 h, (e) 4 h at 1120 °C, and (f) holding at 870 °C for 32 h. After the experiment, the observation plane of the sample was polished. After corrosion with special corrosion solution, the microstructure of the sample was observed under an electron microscope. The results show that the equivalent side length of the precipitated phase under each aging regime, and the specific data are listed in Table 5.

**Table 5.** Side lengths of the precipitated phase under different aging systems.

| Number | (a) | (b) | (c) | (d) | (e) | (f) |
|---|---|---|---|---|---|---|
| Length (nm) | 86 | 187 | 213 | 231 | 242 | 523 |

It can be seen from Figure 7, the equivalent edge lengths of precipitated phase increased with aging time. The precipitated phases of the samples treated only by solid solution showed irregular spherical shapes, and most of them were approximately circular in cross-section. Very few of them began to grow sharp corners where they were close to each other, showing a trend of quadrilateral transformation. However, the distribution of the precipitates was disorderly, only in the initial growth stage after nucleation. After initial growth and merger, a large number of precipitates nuclei formed at the beginning of cooling grow into spherical spheres, and the number decreases was slightly comparable with that at the beginning of cooling. However, due to the growth only in a very short cooling time, the volume was still very small, and it had not grown to the extent that the arrangement could be changed. From a comparison of Figure 8a,b, it can be seen that the precipitates had obvious growth changes after 1 h of aging treatment at 1120 °C. The original spherical precipitation phase grew into a cube shape, which was quadrilateral in cross-section, and the quadrilateral was very obvious, and the cubic degree was greatly improved. Compared with the equivalent side-length of the precipitation phase in Figure 8a, the equivalent side length of the precipitation phase in Figure 8b reaches 187 nm. The volume of the precipitation phase obviously increased, and the number of corresponding phases decreased greatly. At the same time, the distribution of precipitation phases tended to be regular, and the strengthened cubic phases are arranged tightly and orderly. From Figure 8b–e, it can be seen that the precipitates grow and merge further with an increase of the holding time at 1120 °C. The cubic degree of the precipitates increased gradually with the increase of the holding time. The magnification of Figure 8b,e is twice as large as that of Figure 8b,e the size of the mesophase was larger, and the number of mesophases was smaller, and the number of mesophases decreased slightly. On the one hand, the small phase grew gradually because of the continuous migration of elements; on the other hand, the surrounding macrophases also grew continuously. In the process of growing towards each other, the large and small phases broke through the barrier of the matrix phase and merged together, becoming a new and generally larger precipitation phase. After holding for 1 h, the quivalent side length of the phase was 187 nm, and after holding for 4 h, the equivalent side length of the phase reached 242 nm, and the volume of the phase increases obviously. Figure 8f shows that after holding for 4 h at 1120 °C and holding for 32 h at 870 °C the growth of the mesophase has become complete. Compared with the previous cases, the cubic degree is the highest, the arrangement of the mesophase is the most regular, the volume of the phase is the largest, reaching 523 nm, and the size difference between the precipitation phases is reduced, and there is too large or too small. The inclusion reinforcement phase disappears. Compared with the original sample, the size, shape and arrangement of phases are very close. It can be concluded that in the heat treatment system: 1335 4H + 1120 4H/AC + 870 32H/AC phase precipitation can achieve good results.

**Figure 7.** Microstructure of the samples after different aging treatments (**a**) Non-aging, (**b**) Aging system ①, (**c**) Aging system ②, (**d**) Aging system ③, (**e**) Aging system ④, (**f**) Aging system ⑤.

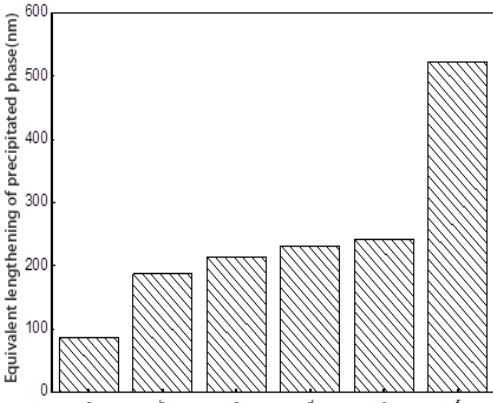

**Figure 8.** Equivalent side lengths of the precipitated phase under different aging treatment conditions.

After solid solution heat treatment and air cooling, the precipitation phase gradually precipitates. In the process of precipitation and growth, various elements, such as Al, migrate, eventually leading to the differences in the content of elements between the precipitation phase and the matrix. From the comparison of Tables 1, 3, and 4, the elemental content in the hardened phase and matrix phase was different from that of the sample after complete heat treatment, only after solid solution treatment. It can be seen that the elements migrate during aging. In order to determine the influence of aging temperature and time conditions on the migration of elements in the precipitated phase and the reinforced phase, an energy spectrum analysis was carried out on the surface of the sample. The principle of selecting points was consistent with the above solid solution treatment. The average elemental contents of each point were obtained at the points of appropriate amount in the reinforced phase and matrix phase, and the chemical composition of the precipitation phase and matrix phase under various aging systems. These are listed in Tables 6 and 7.

**Table 6.** Constituents (mass fraction) of the precipitation phases under three aging systems.

| Aging System | Al | Cr | Co | Ni | Mo | Ta | W | Re |
|---|---|---|---|---|---|---|---|---|
| Non-aging | 3.69 | 5.60 | 10.30 | 60.14 | 3.38 | 5.34 | 9.19 | 2.36 |
| 1120 °C/4 h | 5.01 | 4.55 | 9.23 | 59.26 | 2.57 | 6.54 | 9.29 | 3.55 |
| 1120 °C/4 h, 870 °C/32 h | 5.18 | 4.52 | 9.21 | 61.33 | 0.79 | 7.16 | 7.48 | 3.09 |

**Table 7.** Composition (mass fraction) of the matrix phase under three aging systems.

| Aging System | Al | Cr | Co | Ni | Mo | Ta | W | Re |
|---|---|---|---|---|---|---|---|---|
| Non-aging | 5.18 | 4.66 | 9.49 | 59.21 | 1.67 | 7.32 | 6.36 | 4.13 |
| 1120 °C/4 h | 5.01 | 4.69 | 9.18 | 58.28 | 3.39 | 6.32 | 9.52 | 3.62 |
| 1120 °C/4 h, 870 °C/32 h | 4.64 | 5.21 | 10.57 | 60.67 | 1.73 | 5.31 | 8.74 | 3.13 |

It can be seen from Tables 6 and 7 that during aging, with the precipitation, merger, and growth of the precipitation phase, with $Ni_3Al$ as the main component, Al migrates from the matrix phase to the precipitation phase, with the same migration trend as Ta and Re. It has been mentioned that when the material is treated by a solid solution for enough time to form a single-phase solid solution, during the process of rapid cooling, some elements seem to be more easily attached to the nucleus to form the precipitated phase, and form dendritic structure along a specific growth direction, so that there is usually serious element segregation in the dendritic cadre position. In the subsequent aging process, due to long-term heat preservation at a lower aging temperature, enough growth time was given to the precipitation phase. In the process of growth, the precipitation phase exchanges and merges the elements with the surrounding matrix phase and the fine secondary precipitation phase. The whole system develops towards a more stable structure, and the energy of the system decreases continuously. It can be seen from the above table that elements such as Cr, Co, Mo, and W enriched in the precipitation phase migrate from the precipitation phase to the matrix phase during aging.

## 5. Dissolution and Precipitation Mechanisms

In the dissolution process of the γ′ phase, the elastic field related to the dislocation still exists in the microstructures, although the sample is not affected by external loads, which affects the dissolution behavior of the γ′ phase. Because the channels of cubic reinforcement phase and matrix phase are closely arranged in a chessboard shape, there is elastic interaction between them, thus forming the local stability of the reinforcement phase accumulation group. With the solid solution treatment, the stability of each accumulation group is destroyed, due to the influence of temperature, and the first destroyed accumulation group demonstrates dissolution behavior. Therefore, the overall dissolution behavior of the γ′ phase is not a macroscopic phenomenon that is formed by the simple combination of dissolution of each γ′ phase due, to the influence of temperature. During the solid solution treatment process, some of these γ′ phase stacks connected by elastic interaction become extremely unstable, due to the

absorption of heat energy, and they gradually break away from the bonds between them, and they enter the matrix phase. That is to say, the dissolution behavior occurs, and the original stable structure is destroyed after the loss of this part. The structure of the whole precipitated phase group becomes more unstable. The lack of stability that is caused by partial dissolution destroys the integrity of the elastic strain field, and reduces enough of the elastic strain field. The resistance to $\gamma'$ phase dissolution is weakened, and the dissolution process becomes smoother and more uniform.

In the cooling stage after solid solution treatment, the solubility of the precipitation phase elements in the matrix phase decrease, due to the decrease of temperature, and then nucleation and precipitation begin to grow. In the process of precipitation, the growth behavior of the phase, including the number of nucleations, the size and shape of the precipitation, the direction of precipitation growth, and the precipitation of the secondary precipitate, are all affected by the cooling rate, which determines the nucleation rate and the kinetics of the precipitate growth. Therefore, the morphology, size, and distribution of the precipitated phases and the overall microstructure characteristics can be controlled and described by different cooling rates during the cooling process. In the cooling process, when the cooling rate is low, as in the case of 0.5 °C/s studied previously, due to the slow drop of temperature, low supersaturation in the nucleation stage and slow nucleation of precipitated phase, the number of primary nucleations is small and the nucleation density is low, which makes the precipitated phase of the primary nucleation have enough time to grow and gradually form dendritic structures. When the cooling rate is accelerated, the precipitate phase is "locked" in the early free-growth process.

The microstructures of the specimens after solution treatment were studied by using different cooling rates to reduce the temperature from that of the solid solution to room temperature. When the cooling rate is low, most of the precipitates have undergone relatively full free growth. After a long cooling process, the precipitation phases eventually come close to each other in four groups to form butterfly-like structure knots. The volume of the phase is large and the cube structure with sharp edges and corners is tetragonal in cross-section, and the channel of the matrix phase is relatively wide. When the cooling rate increases, the number of phases increases, the volume of the final phase decreases, and the curvature of the corner decreases, and the channel of the matrix phase becomes finer. When air cooling is used, the precipitation phases in the material show irregular spheres, and the volume of the phase decreases obviously compared with the cooling rate at 0.15 °C. From the cross-section, the phase is irregularly circular, a few have the tendency of edge angle, and the channel of the matrix phase is finer, and there are fine secondary precipitation phases that are precipitated. When the cooling rate is faster than that of water cooling, it is difficult to see the microstructure of the structure under the existing micro-electron microscopy. It can be seen that the micro-spheres are piled up in disorderly manner. Because of the rapid cooling rate, the precipitation phase ends its growth, and solidifies over a very short time after precipitation.

It has been pointed out that the exact shape of the precipitation phase in the cooling process after solution treatment is affected by factors that are related to the elastic field and other diffusion dynamics. In the solid solution treatment process, the precipitation phase and the matrix phase fully melt at a high temperature to form a single-phase solid solution. When the temperature decreases, the solubility of the precipitation phase in the matrix phase decreases, and the solid solution is in the supersaturated state. At this time, the surface shape of the precipitated phase presents a depression shape, although from a purely elastic point of view, the phase is not in a balanced state. When the temperature continues to decrease and the precipitation phase continues to precipitate and grow rapidly from the matrix phase, the metastable precipitation phase blocks are formed next to a dendritic structure that is formed along the growth direction. In the subsequent process, the growth of the precipitate phase makes the surface state of the phase closer to the plane state of the equilibrium state. At this time, precipitated phases continue to precipitate and grow gradually in the decreasing channel of matrix phase, resulting in the continuous increase of the volume fraction of the $\gamma'$ phase. At the end of the cooling stage, it can be observed that there are fine precipitation phases of different sizes in the channel of $\gamma'$ phase, and in the low-saturation region around the primary precipitation

phase. This indicates that during the cooling process, new nuclei form and grow, and the existing precipitation phases continuously absorb various elements from the surrounding matrix phase for growth. The system exists at the same time, and there is competition between them.

## 6. Conclusions

1.  In the process of the solid solution treatment of the recovery heat treatment, as long as the appropriate solution temperature is selected, and after enough time at solid solution treatment, the precipitation phase will fully dissolve in the matrix phase to form uniform single-phase solid solutions. The heating rate has no effect on the structure evolution and element migration of the precipitation phase after cooling.

2.  When the cooling rate is very high, reaching a rate that is similar to air-cooling or water-cooling, the number of primary nucleations of enhanced phase precipitation is very large. The very short cooling time makes the growth degree of the enhanced phase very low, and the shape of enhanced phase is inclined to a irregular sphere, which shows disorder stacking growth. When the cooling rate is about 0.5 °C/s, the reinforcement phase merges and reduces the number of reinforcement phases in the growth process, and it gradually grows into a cube shape, and the distribution tends to be regular. When the cooling rate was about 0.15 °C/s, the precipitation phase grew further, and a butterfly structure consisting of four precipitation phases was formed.

3.  In the aging process, the precipitation phase has enough time to regenerate and grow, and the shape and size of the phase are preliminarily adjusted at a higher aging temperature (1120 °C). From the approximate shape of the sphere after air cooling, it has grown into a more regular cubic structure. Compared with the solid solution treatment, the size of the precipitation phase has increased greatly, and the phase arrangement tends to be compact. The cubic degree of the precipitation phase is further increased at a lower aging temperature (870 °C), and the size of the precipitation phases become close to each other. The inclusion of precipitation phases that are too large or too small gradually disappears, and the phase arrangement is more orderly. The main directions of element migration are as follows: Al, Ta, and Re elements are more concentrated in the matrix phase during the cooling process after solid solution treatment, and they migrate to the precipitation phase during the aging process; the migration direction of the elements Cr, Co, Mo, and W in the aging process is from the precipitation phase to the matrix phase.

**Author Contributions:** Design study, X.-Y.W. and Z.-X.W.; Literature search, S.-N.G. and H.C.; Formal analysis, G.-X.L.; Investigation, S.-N.G. and X.-Y.W.; Data curation, H.C.; Writing—Original draft preparation, X.-Y.W. and H.C.; Writing—Review and editing, X.-Y.W.; Funding acquisition, Z.-X.W.

**Funding:** This research was funded by the National Natural Science Foundation of China, grant number 51875462 and the Innovation Capability Support Plan in Shaanxi Province of China, grant number (2018KJXX-007).

**Acknowledgments:** We are grateful for the financial support provided by the National Natural Science Foundation of China (51875462), the Innovation Capability Support Plan in Shaanxi Province of China (2018KJXX-007).

**Conflicts of Interest:** The authors declare no conflict of interest.

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
