# Peer review of "Influences of the Heating and Cooling Rates on the Dissolution and Precipitation Behavior of a Nickel-Based Single-Crystal Superalloy"

_metals, doi:10.3390/met9030360_

Reviewer 1 Report

In my opinion, this article can be published in the current form. The article is well written and structured. On the other hand, the theme of the presented work is interesting.

Author Response

Thank you for your approval. I will continue to work hard.

Reviewer 2 Report

The submitted paper presents a study on dissolution and precipitation behavior after heat treatment. It has been investigated how the heating rates, cooling rates and ageing temperature change the precipitation behavior of nickel based single crystal superalloy. The experimental process is divided in three stages: solution treatment, cooling and ageing. The behavior of dissolution and precipitation was analyzed with a high power scanning electron microscope. In the solution stage, it was studied if and how heating rate changes the morphology of material. Cooling stage was used to look at cooling rates and alteration of dissolution and precipitation behavior between different cooling rates. In ageing stage, different treatments were used and compared to each other.  At the end, dissolution and precipitations mechanics are described.

Major comments and suggestions to the authors:

The aim of the investigation is not apparent from the paper. Please clarify.

Please check English by native speaker.

Line 120: missing units. Please edit.

Some caption of tables and figures start with capital letters and others don`t. Please change.

Please ensure that the figures and headlines are fir together (see Fig. 4, Fig 6, Fig. 7)

Please clarify is the circular or square shape better in microstructure.

Table 5: “a”, “b”, … are not numbers. Please edit.

Line 273: what is “too large or too small”? Please clarify.

In conclusion is missing if the objective was achieved or not. What for heating rates, cooling rate and ageing treatment combinations are better than standard or is there no recommendation because standard is still better for this material.

Author Response

Thank you for your comments. I will adopt and accept them all and correct them accordingly.

point 1.The aim of the investigation is not apparent from the paper. Please clarify.

The aim of the investigation is influence of heating and cooling rate on dissolution and       precipitation behavior of Nickel-based single crystal superalloy.

From point 2 to point 7, I have corrected it, which will be reflected in the revised paper.

point 8:what is “too large or too small”? Please clarify.

 This is the size of the  γ´phase ,some are too large and some are too small.

In conclusion is missing if the objective was achieved or not. What for heating rates, cooling rate and ageing treatment combinations are better than standard or is there no recommendation because standard is still better for this material.

 The conclusion shows the evolution of the structure and element migration of the γ´phase,the research goal was achieved. This paper is a part of the research on material performance restoration, which will ultimately improve the material performance and prolong the material life through  Rejuvenation treatment, we will continue to describe it in other articles.

 sincerely,

 Wang

Reviewer 3 Report

Line 65 The 65 creep properties were improved at 704 C/690 MPa.

Line 82 Please change "Figure1is" with "Figure 1 is "

Figure 3 Please add a line for Air  Cooled

Line 163 

Line 165 After Figure 5 please use dot.

Line 167 Please xplain that the results are summarized in figure 5 d).

Figure 5 pleae add c) and d).

Line 177 Please cut "the samples are heated from room temperature to target temperature at different heating rates, and after full solution treatment, the temperature is reduced to room temperature at the same 179 cooling rate." and add this sentence in Line 163 after "the specimens."

Author Response

Thank you for your comments. I accept them all and make corresponding corrections in the article.

sincerely,

Wang

Reviewer 4 Report

The aim of the paper, i.e. the study of the effect of various factors (heating rate and cooling rate) on the dissolution and precipitation behavior of γ´ phase and the number, size, morphology and distribution of final precipitates in the heat treatment of nickel-based single crystal superalloy, is quite interesting, even if not so appealing, due to the large number of papers already published on this topic.

The abstract summarize the work. The purpose of the study is clearly outlined and the findings of prior work are well discussed. There are no errors in logic or experimental procedure. However, a part related to etching process and solution treatment requires some small corrections and explanation. The authors accurately explain how the data were collected. There is sufficient information that the experiment can be reproduced. All topics are well presented and discussed. The summary and conclusions are sound and justified. All presented figures are good quality and they prove their point. The paper is written in good English. The manuscript is easily readable concerning language, style and presentation. The references are appropriate and up to date.

Unfortunately, the subject is rather known and widely described. The novelty of this paper should be clearly indicated. Moreover, some experimental assumption like the influence of heating rate on the structure after solution process doesn’t make any sense.

Following are some of my comments.

1. Line 78

It is “2. Test materials and Methods”

I suggest “2. Materials and Methods”

2. Line 79

It Is “Test Material”

I suggest “Material”

3. Line 89

It is “Test Sample”

I suggest “Sample test”

4. Lines 97 and 98

It is “Chemical corrosion solution”

I suggest “Chemical etching solution”

5. Lines 98 and 99

It is “The corrosive droplets”

I suggest “The etching droplets”

6. Line 100

It is “the corrosion operation”

I suggest “the etching operation”

7. Lines 120-122

Authors wrote “In order to simulate the condition of rapid heating, the sample was pre-heated to 1335 °C in the high temperature furnace, and then quickly placed in the high temperature furnace, so that the sample could be heated to 1335 °C in the shortest time.” In my opinion something is wrong with pre-heating and final temperatures. Are they equal?

8. Fig. 4 No units at axis descriptions (time and temperature)

9. Paragraph 4.1 In the lines 126-130 the authors wrote “Because the temperature in the early stage of the heating process is very low and far below the dissolution temperature of the precipitation phase, many precipitation phases remain undissolved at the end of the heating process, and because the different heating rates lead to the different amount of dissolution of the precipitation phase at the end of the heating process, the uniform holding of the sample at 1335 °C for 4 h is chosen to ensure the complete dissolution of the phase, and then the air cooling is carried out.” This means that any influence of the heating rate on the phase precipitation process is overcome by a long annealing time and a further study on the influence of the heating rate does not make sense. I suggest to remove form the paper the whole part related to the material obtained using various heating rate.

Author Response

Thank you for your comments and suggestions. I accept them all and make corresponding corrections in the article.

 From point 1 to point 6,point 8, I have corrected them, which will be reflected in the revised paper.

 for point 7

There's a mistake in this part. I've removed it.

for point 9.

The heating rate has no effect on the structure evolution and element migration of the strengthened phase.I did a lot of experimental work in the early stage. I want to remove part and leave part. Is that OK?

Reviewer 5 Report

Dear Authors,

the article is quite good, but I have some comments and suggestions to address. After you do it I will recommend the article for publishing.

1.       Key words are too extended. Greek symbol “gamma” is impossible to type in the international databases.

2.       Line 23 and 66: please decide: aeroengine or aero-engine.

3.       How many samples were analyzed? I am afraid not too many, looking into the results. Are the results representative and statistically valuable?

4.       Most graphics must be improved, especially lettering (to small thus invisible)

5.       Line 146: what kind of process is presented in the Figure 3?

6.       Please decide if “aging” or “ageing” term is used in the article

7.       Line 238: incomplete caption under the Figure 7

8.       Please decide the symbol for “hours”: “h” or “H”

Sincerely,

Reviewer

Author Response

Thank you for your comments and suggestions. I accept them all and make corresponding corrections in the article.

1.for point 1 , point 2,from point 4 to point 8,I have corrected them, which will be reflected in the revised paper.

for point 3,How many samples were analyzed? I am afraid not too many, looking into the results. Are the results representative and statistically valuable?

For the same sample, different heating and cooling rates, in fact, each experiment has been done at least three times, which is not reflected in the paper,An explanation of this point has been added to the paper.the results are representative and statistically valuable.

Sincerely,

Wang

Round  2

Reviewer 4 Report

The manuscript has been properly revised.

Author Response

I have checked and revised it according to the grammatical rules and styles of the English language.

Thank you.